# γ-Oryzanol from Rice Bran Antagonizes Glutamate-Induced Excitotoxicity in an In Vitro Model of Differentiated HT-22 Cells

**DOI:** 10.3390/nu16081237

**Published:** 2024-04-21

**Authors:** Li-Chai Chen, Mei-Chou Lai, Tang-Yao Hong, I-Min Liu

**Affiliations:** 1Department of Pharmacy and Master Program, Collage of Pharmacy and Health Care, Tajen University, Pingtung County 90741, Taiwan; icupdrab@tajen.edu.tw (L.-C.C.); meei@tajen.edu.tw (M.-C.L.); 2Department of Environmental Science and Occupational Safety and Hygiene, Graduate School of Environmental Management, Collage of Pharmacy and Health Care, Tajen University, Pingtung County 90741, Taiwan; tyhong@tajen.edu.tw

**Keywords:** glutamate, vascular dementia, γ-oryzanol, differentiated HT-22 cells, neurodegenerative diseases

## Abstract

The excessive activation of glutamate in the brain is a factor in the development of vascular dementia. γ-Oryzanol is a natural compound that has been shown to enhance brain function, but more research is needed to determine its potential as a treatment for vascular dementia. This study investigated if γ-oryzanol can delay or improve glutamate neurotoxicity in an in vitro model of differentiated HT-22 cells and explored its neuroprotective mechanisms. The differentiated HT-22 cells were treated with 0.1 mmol/L glutamate for 24 h then given γ-oryzanol at appropriate concentrations or memantine (10 µmol/L) for another 24 h. Glutamate produced reactive oxygen species and depleted glutathione in the cells, which reduced their viability. Mitochondrial dysfunction was also observed, including the inhibition of mitochondrial respiratory chain complex I activity, the collapse of mitochondrial transmembrane potential, and the reduction of intracellular ATP levels in the HT-22 cells. Calcium influx triggered by glutamate subsequently activated type II calcium/calmodulin-dependent protein kinase (CaMKII) in the HT-22 cells. The activation of CaMKII-ASK1-JNK MAP kinase cascade, decreased Bcl-2/Bax ratio, and increased Apaf-1-dependent caspase-9 activation were also observed due to glutamate induction, which were associated with increased DNA fragmentation. These events were attenuated when the cells were treated with γ-oryzanol (0.4 mmol/L) or the N-methyl-D-aspartate receptor antagonist memantine. The results suggest that γ-oryzanol has potent neuroprotective properties against glutamate excitotoxicity in differentiated HT-22 cells. Therefore, γ-oryzanol could be a promising candidate for the development of therapies for glutamate excitotoxicity-associated neurodegenerative diseases, including vascular dementia.

## 1. Introduction

Vascular dementia is the second most prevalent type of dementia, next to Alzheimer’s disease. It happens when blood flow to the brain is restricted due to a series of small strokes or other vascular issues that affect the blood vessels that supply the brain with nutrients [1]. In addition to strokes, risk factors for vascular dementia include advanced age, diabetes, hypertension, and atherosclerosis [1]. The main symptom of vascular dementia is cognitive decline, including memory loss, difficulty with problem-solving and planning, confusion, and changes in behavior and mood [2]. Vascular dementia may have a different progression compared to Alzheimer’s disease; there may be more sudden declines in symptoms after strokes or vascular events rather than gradual worsening [2]. The hippocampus is a vital brain region responsible for learning, memory processing, and cognition [3]. Research has revealed a close association between the hippocampus and its blood vessels and brain injury caused by transient cerebral ischemia; this highlights the importance of this area and its potential role as a target for tackling dementia [4]. Effective management of vascular dementia involves addressing underlying vascular risk factors and promoting brain health. Primary preventive medications, such as cholinesterase inhibitors and excitatory amino acid receptor antagonists, have limited effectiveness in treating vascular cognitive impairment [5]. At present, there are no curative treatments available for vascular dementia, which highlights the urgent and unmet need for advancements in therapeutic options [5]. 

Understanding the molecular mechanisms underlying these processes is crucial for developing therapeutic interventions to prevent or slow down vascular dementia [5]. While the exact causes of vascular dementia are not fully understood, research using different experimental models of cerebral ischemia suggests that glutamate-induced excitotoxicity is strongly associated with the development of the condition [6]. In the central nervous system, glutamate is an essential neurotransmitter that stimulates neurons, impacting synaptic transmission, synaptic plasticity, and cognitive function in a significant way [7]. In ischemic-hypoxic conditions, excessive release of glutamate in the synaptic cleft causes calcium influx through N-methyl-D-aspartate (NMDA)-subtype glutamate receptors, triggering mitochondrial permeability transition pores, and as a result, initiating the mitochondria-dependent apoptosis [8]. The apoptosis of neuron cells induced by glutamate occurs not only through the mitochondria-dependent apoptosis pathway. The c-Jun N-terminal kinase (JNK) is a member of the mitogen-activated protein kinase (MAPK) family, which in turn phosphorylates c-Jun to regulate the transcription of AP-1 target genes involved in the apoptotic pathway downstream [9]. Excitotoxicity and being under conditions of excessive NMDA receptor activation can activate the JNK/c-Jun/AP-1 signal transduction pathway, further amplifying apoptotic signaling in ischemic cell death [10]. Research focuses on developing pharmaceuticals that selectively target NMDA receptors or the JNK signaling pathway, aiming to counteract the detrimental effects of glutamate excitotoxicity, which has been shown as a potential therapeutic strategy for vascular dementia [6].

γ-Oryzanol (9,19-Cyclo-9β-lanost-24-en-3β-ol 4-hydroxy-3-methoxycinamate) is a naturally occurring compound that is abundant in rice bran oil consisting of ferulic acid esters and phytosterols [11]. The nutrition and health benefits of γ-oryzanol have sparked interest, specifically its antioxidant properties that help to counteract harmful free radicals in the body [12]. Additionally, γ-oryzanol has been researched for its potential anti-inflammatory, anti-cancer, anti-hyperlipidemic, anti-ulcerogenic, anti-diabetic, and hormone-balancing effects [11]. For athletes and those involved in sports, γ-oryzanol may also have a positive impact on muscle growth and recovery [13]. Studies have shown that γ-oryzanol can pass through the blood–brain barrier (BBB) without breaking down and is widely distributed within the brain [14]. In a rat model of sporadic Alzheimer’s disease caused by streptozotocin, γ-oryzanol has shown potential in preserving cognitive function and memory [15]. γ-Oryzanol has also been found to improve working and long-term memory in adult mice by regulating changes in the hippocampal proteome [16]. However, the specific effects of γ-oryzanol on hippocampal cells and its potential in treating glutamate-induced excitotoxicity, a process strongly associated with the development of vascular dementia, have not been extensively researched.

Comparing cellular and animal models, the former offers several benefits, including quicker development of pathology, cost savings, and exemption from ethical approval requirements [17]. Cellular models thus aid in identifying drug targets and can be validated through animal testing [17]. The hippocampal structure in the brain is the core division for spatial cognitive activity [18]. HT-22 cells derived from hippocampal neurons, once differentiated, exhibit NMDA receptor expression and become more sensitive to excitotoxicity [19,20]. Therefore, differentiated HT-22 cells are susceptible to serving as mature hippocampal neurons in in vivo models for studying neurodegenerative diseases associated with glutamate-induced excitotoxicity [19,20]. This study investigates whether γ-oryzanol has therapeutic potential to delay or improve vascular dementia in an in vitro cell model of differentiated HT-22 cells under glutamate-induced excitotoxicity and explores the molecular mechanisms involved in its neuroprotective properties. Memantine is a widely recognized medication that functions as a partial antagonist of the NMDA receptor, thus helping restore the equilibrium of the glutamatergic system, leading to enhanced cognitive function and memory [21]. The study utilized memantine as a standard compound to compare and assess the impact of γ-oryzanol. These results could serve as a reference for γ-oryzanol as a potential therapy for neurodegenerative diseases associated with glutamate excitotoxicity.

## 2. Materials and Methods

### 2.1. Cell Culture

The immortalized mouse hippocampal neuronal HT-22 cells were purchased from the American Type Culture Collection (Manassas, VA, USA) and cultured until reaching 80% confluence in Dulbecco’s modified Eagle’s medium (DMEM). The growth medium was supplemented with a 1% antibiotic mixture containing 100 U/mL penicillin and 100 μg/mL streptomycin, 10% fetal bovine serum (FBS), and L-glutamine. Cultures were maintained in a humidified atmosphere at 37 °C with 5% CO_2_. The cells were subjected to differentiation in Neurobasal™-A medium supplemented with 2 mmol/L L-glutamine, 1 × N2 supplement (Life Technologies, Carlsbad, CA, USA), and a 1% antibiotic mixture for 48 h prior to treatments [19,20]. Differentiation and neurite growth were captured using an inverted microscope (Zeiss Axiovert 135, Carl Zeiss GmbH, One Zeiss Drive, Thornwood, NY, USA) with an XY motorized stage and “mark and find” module of AxioVision (version 4.8.1) image analysis software (Carl Zeiss GmbH, One Zeiss Drive, Thornwood, NY, USA). The number of neurites originating directly from the soma were manually counted to determine the total number of neurites per cell, with analysis performed using Image J software (version 1.6.0, Wayne Rasband, National Institutes of Health, Bethesda, MD, USA). After the differentiation process, cells were seeded at a density of 1 × 10^5^ in each well of 6-well plates. Cultures were dissociated using 0.05% (*w*/*v*) trypsin in phosphate-buffered saline (PBS) with a pH of 7.4 upon reaching confluence.

### 2.2. Procedure of Glutamate-Induced Excitotoxicity and Treatment

To investigate the benefit role of test compounds on the glutamate-induced excitotoxicity, differentiated HT-22 cells were treated with 0.1 mmol/L glutamate for 24 h and then incubated with either γ-oryzanol (Sigma-Aldrich, St. Louis, MO, USA; Cat. # CDS021604) at specified concentrations (ranging from 0.2 to 0.4 mmol/L) or 10 µmol/L memantine hydrochloride (Sigma-Aldrich, St. Louis, MO, USA; Cat. # M9292) for another 24 h at 37 °C [22]. Differentiated HT-22 cells that were not subjected to any treatment were used as control samples. To prepare the experimental compounds, a stock solution with a concentration of 1 mmol/L was made using dimethyl sulfoxide (DMSO, Sigma-Aldrich, St. Louis, MO, USA; Cat. # D8418) as the solvent. The stock solution was then appropriately diluted in the culture medium to obtain concentrations suitable for the subsequent study. The final concentration of DMSO was 0.1% (*v*/*v*), a concentration known to be non-toxic to cells [23].

### 2.3. Assement of Cell Viability

The Cell Counting Kit-8 (CCK-8 kit, Cat. 96992) was used to measure cell viability according to the manufacturer’s instructions from Sigma-Aldrich (St. Louis, MO, USA). HT-22 cells were seeded in 96-well plates at 2 × 10^4^ cells/mL density. After the designated treatments, each well received 10 μL of CCK-8 reagent, followed by a 4 h incubation at 37 °C. The optical density of each well at 450 nm was assessed employing a multifunction microplate reader (SpectraMax M5, Molecular Devices, Sunnyvale, CA, USA). The control cells treated with the vehicle alone were considered 100% viable, and the cell viability of the treated samples was expressed as a percentage relative to the control.

### 2.4. Measurement of Intracellular Reactive Oxygen Species (ROS)

The concentration of dichloro-dihydro-fluorescein diacetate (DCFH-DA) was adjusted to 10 μmol/L by diluting it with DMEM for the assessment of cellular ROS levels [24]. Following a 30 min incubation period at 37 °C with DCFH-DA, the cells underwent three washes with PBS. The fluorescence intensity of dichlorofluorescein was measured using a SpectraMax M5 multifunctional microplate reader (Molecular Devices, Sunnyvale, CA, USA) with excitation at 488 nm and fluorescence emission detection at 525 nm. ROS levels were indicated as a percentage relative to the baseline ROS level observed in untreated cells or cells maintained under normal conditions.

### 2.5. Measurement of Antioxidant Enzymes Activity and Intracellular Glutathione Level

Enzyme-linked immunosorbent assay (ELISA) commercial kits were used to estimate the antioxidant activities. The Activity Colorimetric Assay Kits of superoxide dismutase (SOD; Cat. #ab65354), glutathione peroxidase (GSH-Px; Cat. #ab102530), and catalase (CAT; Cat. #ab83464) were obtained from Abcam plc (Cambridge, MA, USA). The activities of SOD, GSH-Px, and CAT were assessed by measuring absorbance at 450 nm, 340 nm, and 570 nm, respectively, using a microplate reader (SpectraMax M5, Molecular Devices, Sunnyvale, CA, USA). Enzymatic activities were normalized and expressed in units per mg of protein, with protein concentration determined to facilitate normalization.

The glutathione colorimetric assay kit (Abcam plc, Cambridge, MA, USA; Cat. # ab239709) was utilized to measure cellular glutathione (GSH) levels, involving the reduction of 5,5′-dithiobis-(2-nitrobenzoic) acid to 2-nitro-5-thiobenzoic acid, with measurements taken at 405 nm. The concentration of GSH is expressed in nmol per mg of protein. Protein quantification was performed using a Bradford protein assay [25].

### 2.6. Measurement of Intracellular Calcium

To determine intracellular calcium concentration, the fluorescent dye Fluo-3/AM (Sigma-Aldrich, St. Louis, MO, USA; Cat. # 343242-M) was utilized [26]. The cells underwent a 30 min incubation with a 3 μmol/L Fluo-3/AM working solution in darkness at 37 °C, followed by washing with PBS to eliminate any unbound dye. Calcium images were captured using a fluorescence microscope (Leica Microsystems, Wetzlar, Hessen, Germany). The fluorescence intensity of calcium was assessed utilizing a microplate reader, with excitation occurring at a wavelength of 488 nm and emission measured at 525 nm. Quantitative assessment of the images was performed utilizing Leica image analysis software (version 3.7). This software computed the fluorescence intensity within the designated region of interest.

### 2.7. Assessment of Mitochondrial Transmembrane Potential

The JC-1 dye from Abcam plc, Cambridge, MA, USA (Cat. # ab113850) was used to assess mitochondrial membrane potential. The cells were seeded into a 96-well plate at a density of 1 × 10^4^ cells/well and then incubated with 20 µmol/L JC-1 at 37 °C for 30 min. After incubation, the cells were centrifuged at 2500 rpm for 5 min, and the resulting pellets were resuspended in 0.5 mL of PBS. The fluorescence spectrophotometer (SpectraMax M5, Molecular Devices, Sunnyvale, CA, USA) was used to quantify the red aggregates emitting at 590 nm and the green-fluorescent monomers with a 530 nm emission. The mitochondrial transmembrane potential was evaluated by calculating the ratio of J-aggregates to J-monomers [27].

### 2.8. Quantification of ADP and ATP Concentrations

The ADP/ATP Ratio Assay Kit (Cat. #ELDT-100) from BioAssay Systems employs luciferase to generate light when its luciferin substrate is present [28]. Cells were lysed using 10% trichloroacetic acid, neutralized with 1 mol/L KOH, and diluted with 100 mmol/L HEPES buffer (pH 7.4) after treatment. In the initial stage of the assay, cellular ATP and D-luciferin underwent a luciferase-catalyzed reaction, resulting in a luminescent signal. ADP was converted to ATP by enzymatic activity, followed by reaction with D-luciferin to produce a luminescent signal proportional to ADP and ATP amounts. The normalization of the ADP/ATP ratio was performed with respect to the total protein content present in the samples.

### 2.9. Quantification of Released Cytochrome C

Following treatment, the cells underwent homogenization, and the resulting lysate was centrifuged at 800× *g* for 20 min. The resultant supernatant was then further centrifuged at 10,000× *g* for 15 min to isolate the mitochondrial fraction. The remaining supernatant underwent a subsequent centrifugation at 16,000× *g* for 25 min to obtain the cytosolic fraction. The levels of cytochrome c in both fractions were determined using the cytochrome C ELISA kit (Abcam plc, Cambridge, MA, USA; Cat. #ab110172), following the manufacturer’s instructions. Cytochrome c was captured within the wells by a specific antibody conjugated with horseradish peroxidase. The peroxidase catalyzed a color change in the substrate from colorless to blue, which was then neutralized by the addition of 100 μL of 1.5 N HCl to each well, and the resulting solution was measured at 450 nm. The protein content was quantified using Bio-Rad protein analysis [25].

### 2.10. Measurement of Mitochondrial Complex I Activity

The activity of mitochondrial complex I was assessed utilizing an ELISA kit (Cat. #AB109721) in accordance with the manufacturer’s guidelines provided by Abcam plc (Cambridge, MA, USA). After lysing the cells with a detergent solution, the incubation solution was adjusted to 1 mg/mL protein concentration. The 200 μL sample and control were applied to a pre-coated microplate with complex I capture antibody, followed by 3 h incubation at room temperature. The activity of complex I was determined by spectrophotometric measurement of the oxidation of NADH to NAD+ and concurrent reduction of a dye at 450 nm. The activity was expressed as nmol of oxidized NADH per minute per milligram of protein, with protein levels quantified using Bio-Rad protein analysis [25].

### 2.11. Quantification of Apoptotic DNA Fragmentation

DNA fragment quantification linked with cytoplasmic histones stemming from induced cell death was conducted utilizing the cell death detection ELISA kit (Roche Molecular Biochemicals, Mannheim, Germany; Cat. #11774425001). Cytoplasmic cell extracts were utilized as the antigen source in a sandwich ELISA configuration, wherein a primary mouse monoclonal antibody against histones was immobilized onto the microtiter plate, followed by the application of a secondary mouse monoclonal antibody against DNA, conjugated to peroxidase. The photometric determination of the peroxidase retained within the immune complex was conducted by incubating it with the substrate 2,2′-azino-di-[3-ethylbenzthiazoline sulfonate] for 10 min at 20 °C. The subsequent alteration in color was assessed using a microplate reader (SpectraMax M5, Molecular Devices, Sunnyvale, CA, USA) at a wavelength of 405 nm.

### 2.12. Western Blot Analysis

Cells were harvested and lysed in ice-cold radioimmunoprecipitation buffer for 30 min. The Bradford method was used to determine protein concentration, and 50 μg of protein from each sample underwent Western blot analysis. Protein separation was carried out on a 10% sodium dodecyl sulfate-polyacrylamide gel, followed by electrophoretic transfer onto polyvinylidene difluoride membranes. After blocking with 5% non-fat dry milk in tris-buffered saline containing 0.1% Tween for 3 h at room temperature, the membranes were incubated overnight at 4 °C with primary antibodies targeting NMDA receptor 1 (NMDAR1; Cat. #5704), p-NMDAR1 (Ser890) (Cat. #3381), calcium/calmodulin-dependent protein kinase II (CaMKII, Cat. #3362), p-CaMKII (Thr286) (Cat. #12716), ASK1 (Cat. #8662), p-ASK1 (Ser967) (Cat. #3764), MKK4 (Cat. #9152), p-MKK4 (Ser257/Thr261) (Cat. #9156), MKK7 (Cat. #4172), p-MKK7 (Ser271/Thr275) (Cat. #4171), JNK (Cat. #9252), p-JNK (Thr 183/Tyr 185) (Cat. #9251), c-Jun (Cat. #9162), p-c-Jun (Ser73) (Cat. #9164), c-Fos (Cat. #4384), p-c-Fos (Ser32) (Cat. #5348), Apaf-1(Cat. #5088), caspases-9 (Cat. #9502), cleaved caspases-9 (Asp353) (Cat. #9509), caspases-3 (Cat. #9662), cleaved caspases-3 (Asp175) (Cat. #9661), PARP (Cat. #9542), cleaved PARP (Asp214) (Cat. #9544), Bcl-2 (Cat. # 2876), Bax (Cat. #2772), or β-actin (Cat. #4967). The antibodies, all obtained from Cell Signaling Technology, Inc. (Danvers, MA, USA), were diluted at 1:1000. Following washing with tris-buffered saline containing 0.1% Tween^®^ 20 detergent, the blots were exposed to secondary antibodies at room temperature for 1.5 h before visualization using chemiluminescence (Amersham Biosciences, Amersham, UK). The band densities were quantified using ATTO Densitograph Software (version 4; ATTO Corporation, Tokyo, Japan), and the results were normalized to the β-actin level. All values were normalized by setting the density of untreated control samples to 1.0, indicating a “fold change” in expression differences. Data were obtained from five independent experiments.

### 2.13. Statistical Analysis

The data are presented as the mean ± standard deviation (SD). The statistical analysis was conducted using SigmaPlot Version 14.0 software (Systat Software Inc., San Jose, CA, USA). Each experiment included testing three wells, and the experiment was repeated at least five times. To assess significant differences from the vehicle controls, one-way ANOVA analysis was used, followed by Dunnett’s test as a post hoc analysis. The statistical significance level was set at *p* < 0.05.

## 3. Results

### 3.1. Differentiation Induces Morphological Changes and an Increase in NMDA Receptor Expression in HT-22 Cells

Before differentiation, the cells had a more rounded morphology with few neurites or apparent synapses (Figure 1A, left panel). After being cultured for 48 h in a differentiation medium, the immature cells transformed into a neuron-like triangular shape, and some of these cells developed extended neurites (Figure 1A, right panel). The expression of NMDA receptors in HT-22 cells increased by 1.6-fold after 48 h of differentiation medium incubation (Figure 1B).

### 3.2. γ-Oryzanol Alleviates Glutamate-Induced Neuronal Insults

Changes in cell morphology before and after treatment are depicted in Figure 2A. γ-Oryzanol exhibited a concentration-dependent effect on the increase in cell viability reduced by 0.1 mmol/L glutamate (Figure 2B). Neurite outgrowth induced by 0.1 mmol/L glutamate was reduced to 47.6% of the control group but was elevated in a concentration-dependent manner by γ-oryzanol (Figure 2C). The concentration of γ-oryzanol at 0.4 mmol/L was chosen for subsequent experiments due to its significant effect similar to that produced by memantine (10 µmol/L). While γ-oryzanol (0.4 mmol/L) did not impact the increase in NMDAR1 protein expression induced by glutamate, it did prevent the excitotoxic effects of increasing the phosphorylation level of NMDAR1 (Figure 2D). Memantine (10 µmol/L) eliminated higher NMDAR1 phosphorylation but not its protein level from glutamate (Figure 2D).

### 3.3. γ-Oryzanol Alleviates Glutamate-Induced Oxidative Stress

In the differentiated HT-22 cells, glutamate caused an increase in intracellular ROS levels to 2.5-fold of the untreated control (Figure 3A). When HT-22 cells were treated with γ-oryzanol (0.4 mmol/L) or memantine (10 µmol/L), glutamate-induced ROS production decreased by 36.5% and 39.3%, respectively (Figure 3A).

When the differentiated HT-22 cells were treated with 0.4 mmol/L γ-oryzanol, the intracellular activities of SOD, GSH-Px, and CAT, which had lowered due to glutamate, increased to 2.8-, 2.9-, and 2.3-fold, respectively (Figure 3B). The group that was treated with memantine (10 µmol/L) also exhibited elevated levels of SOD, GSH-Px, and CAT activities, which were reduced by glutamate (Figure 3B, upper panel).

In the differentiated HT-22 cells cultured with glutamate, the concentration of GSH was found to be 72.5% lower when compared to the normal group (Figure 3B, lower). However, when treated HT-22 cells with either γ-oryzanol (0.4 µmol/L) or memantine (10 µmol/L), the GSH levels were increased to 2.6- and 2.8-fold of their respective levels in the vehicle-treated group (Figure 3B, lower panel).

### 3.4. γ-Oryzanol Suppressed Glutamate-Induced Intracellular Calcium Overload and CaMKII Activation

After exposing the differentiated HT-22 cells to glutamate, there was a significant rise in the fluorescence intensity of Fluo-3/AM as compared to the control group. Treatment cells with γ-oryzanol (0.4 µmol/L) or memantine (10 µmol/L) decreased the induced elevation of calcium fluorescence intensity caused by glutamate (Figure 4A).

Glutamate exposure increased CaMKII phosphorylation by 2.4-fold in HT-22 cells. Treatment with γ-oryzanol (0.4 µmol/L) or memantine (10 µmol/L) reduced the elevated phosphorylation in CaMKII by 27.1% and 23.6%, respectively (Figure 4B). The protein levels of CaMKII were not influenced by γ-oryzanol (0.4 µmol/L) or memantine (10 µmol/L) (Figure 4B).

### 3.5. γ-Oryzanol Attenuates Glutamate-Induced Mitochondrial Dysfunction

The mitochondrial membrane potential of the differentiated HT-22 cells exposed to glutamate decreased to 37.2% of the untreated control cells, but treatment with γ-oryzanol or memantine reduced the reduction (Figure 5A).

In the differentiated HT-22 cells stimulated with glutamate, the ADP/ATP ratio was 2.3-fold higher than untreated controls; however, there was a reduction of 39.2% when treated with γ-oryzanol (0.4 µmol/L), as evidenced by Figure 5B. Treatment cells with memantine (10 µmol/L) resulted in the ADP/ATP ratio less than 41.2% of the glutamate induction values, as shown in Figure 5B.

The presence of glutamate resulted in a 58.6% reduction in the activity of mitochondrial chain I complexes in the differentiated HT-22 cells compared to the untreated control. This effect was lessened when the cell was treated with 0.4 µmol/L γ-oryzanol or 10 μmol/L memantine, with a resulting activity of 82.1% and 87.5% of the untreated control (Figure 5C).

### 3.6. γ-Oryzanol Downregulates ASK1/JNK Signaling Pathway Induced by Glutamate

Glutamate caused a 2.8-fold increase in ASK1 phosphorylation in the differentiated HT-22 cells, but this was reduced by 30.6% with γ-oryzanol and 34.8% with memantine treatment (Figure 6A).

Treatment cells with γ-oryzanol (0.4 µmol/L) resulted in a decrease in glutamate-induced phosphorylation of MKK4 and MKK7, with values 27.5% and 24.8% lower than the glutamate group (Figure 6B,C). Similarly, when cells were treated with memantine (10 μmol/L), the glutamate-induced phosphorylation of MKK4 and MKK7 was decreased by 32.3% and 29.6%, respectively (Figure 6B,C).

The differentiated HT-22 cells that received glutamate stimulation showed a 2.8-fold increase in JNK phosphorylation compared to the non-stimulated cells (Figure 6D). The group treated with γ-oryzanol (0.4 µmol/L) or memantine (10 µmol/L) displayed a reduced increase, with values of 22.8% and 27.0%, respectively, when compared to glutamate induction (Figure 6D).

The results showed that exposure to glutamate led to a significant increase of 2.9-fold in the phosphorylation of c-Jun (Figure 6E) and 3.1-fold in the phosphorylation of c-Fos (Figure 6F). However, when the differentiated HT-22 cells were treated with γ-oryzanol (0.4 µmol/L) after the glutamate exposure, the phosphorylation of c-Jun and c-Fos was reduced by 27.5% and 21.4%, respectively (Figure 6E,F). Moreover, treatment of cells with memantine (10 μmol/L) reduced 21.6% in c-Jun phosphorylation and 26.5% in c-Fos phosphorylation induced by glutamate (Figure 6E,F). 

The protein levels of ASK1, MKK4/7, JNK, c-Jun, or c-Fos are not affected by either γ-oryzanol or memantine (Figure 6). 

### 3.7. γ-Oryzanol Declined Cytochrome c-Initiated Caspase Cascade Activation Induced by Glutamate

When exposed to glutamate, the differentiated HT-22 cells experienced a decline in mitochondrial cytochrome c levels and a rise in cytosolic cytochrome c levels (Figure 7A). Treatment with γ-oryzanol (0.4 µmol/L) or memantine (10 µmol/L) helped lessen the release of cytochrome c from mitochondria to the cytoplasm in the differentiated HT-22 cells caused by glutamate (Figure 7A).

The presence of glutamate resulted in a 3.4-fold increase in the protein expression of Apaf-1, as shown in Figure 7B. However, the group that was treated with γ-oryzanol (0.4 µmol/L) or memantine (10 µmol/L) had lower values of 26.4% and 32.4%, respectively, compared to the glutamate-induced group (Figure 7B).

In the differentiated HT-22 cells that were stimulated with glutamate, the protein levels of cleaved caspase-9 and -3 were found to be 3.7- and 3.1-fold higher as compared to those in non-stimulated cells (Figure 7B). However, when the cells were treated with either γ-oryzanol (0.4 µmol/L) or memantine (10 µmol/L) after glutamate induction, the levels of cleaved caspase-9 protein expression were reduced by 44.1% and 39.6%, respectively (Figure 7B). Similarly, the high levels of cleaved caspase-3 in cells that were stimulated with glutamate were reduced to 49.3% and 52.6%, respectively, in cells that were treated with γ-oryzanol (0.4 µmol/L) or memantine (10 µmol/L; Figure 7B).

The levels of cleaved PARP protein in the differentiated HT-22 cells exposed to glutamate was 3.6-fold higher than in the control group; treatment with γ-oryzanol (0.4 µmol/L) or memantine (10 µmol/L) reduced the levels by 42.7%, and 45.8%, respectively (Figure 7B).

### 3.8. γ-Oryzanol Reversed Bcl-2 Protein Downregulation and DNA Fragmentation Induced by Glutamate

In Figure 8A, it was observed that glutamate reduced the amount of Bcl-2 protein while simultaneously increasing the amount of Bax protein, resulting in a decrease in the Bcl-2/Bax ratio in the differentiated HT-22 cells. However, when the differentiated HT-22 cells were treated with either γ-oryzanol (0.4 µmol/L) or memantine (10 µmol/L), the decrease in Bcl-2 protein and increase in Bax protein caused by glutamate were inhibited, leading to an increase in the Bcl-2/Bax ratio (Figure 8A).

Glutamate caused a 2.3-fold increase in apoptotic DNA fragmentation in the differentiated HT-22 cells, but treatment with γ-oryzanol or memantine reduced it by 43.2% and 42.2%, respectively (Figure 8B).

## 4. Discussion

Neurodegenerative diseases can be caused by high levels of glutamate, leading to neuronal death under certain conditions such as ischemia/reperfusion injury [29]. A potential solution to the glutamate excitotoxicity could be a promising candidate for combating neurodegenerative processes [29]. HT-22 cells, once differentiated, exhibit neuronal traits such as neurite outgrowth and NMDA receptor expression, while undifferentiated cells do not [19,20]. Differentiated HT-22 cells thus become more glutamate-receptive and excitatory [19]. Undifferentiated HT-22 cells respond to glutamate-induced cytotoxicity at higher concentrations of 2.5 mmol/L [19]. HT-22 cells can be damaged with glutamate concentrations as low as 0.1 mmol/L under experimental conditions, which suggests that we used differentiated HT-22 neurons [19]. To investigate glutamate-mediated excitotoxicity, the study employed differentiated HT-22 hippocampal neuronal cells.

Glutamate acts as a substrate for complex I of the mitochondrial electron transport chain through the glutamate–malate shuttle [30]. It is, therefore, that excess glutamate could increase electron flow into the transport chain, leading to elevated ROS production [31]. A balance between ROS production and antioxidant defense systems, including enzymatic antioxidants and other non-enzymatic antioxidants, is critical for cellular homeostasis and preventing oxidative stress-induced apoptosis [32]. The reduction in SOD, GSH-Px, and CAT activity under glutamate-induced oxidative stress highlights the vulnerability of HT-22 cells to neurotoxic insults and emphasizes the importance of maintaining a robust antioxidant defense system to protect against oxidative damage [31]. Glutamate exposure not only leads to impairment of the major antioxidant enzymes but also causes a reduction in intracellular levels of glutathione (GSH), a cysteine-containing tripeptide against oxidative stress [33]. According to the study, γ-oryzanol can help alleviate the increase in ROS and decrease in the activities of SOD, GSH-Px, and CAT, as well as GSH depletion in glutamate-exposed HT-22 cells. The results suggest that the beneficial effect of γ-oryzanol against glutamate-induced neurotoxicity may be due to its ability to enhance the first-line antioxidant defense by scavenging different radicals, thereby reducing oxidative damage. Supportive results emerged from a Drosophila melanogaster model of Parkinson’s disease induced by rotenone, affirming that γ-oryzanol enhanced antioxidant defenses, effectively mitigating oxidative stress [34].

ROS has been found to lower the levels of endogenous Bcl-2 within cells [35]. Reducing levels of Bcl-2 could sensitize cells to apoptosis, as Bcl-2 is critical for preventing complete apoptosis [36]. In contrast, Bax regulates the outer mitochondrial membrane’s permeability, releasing cytochrome c from mitochondria [35]. Cytochrome c plays a crucial role in caspase activation by interacting with Apaf-1 in the cytosol, forming the apoptosome, which subsequently activates downstream effector caspases, particularly caspase 9, initiating apoptosis [37]. Moreover, activated caspase-3 cleaves PARP-1, suppressing DNA repair mechanisms and facilitating caspase-mediated DNA fragmentation, observed in various neurological diseases [38]. Our study reveals an evident occurrence of mitochondrial-mediated apoptosis in HT-22 cells exposed to glutamate, including a decrease in the mitochondrial membrane potential, a reduction in ATP levels, a lower ratio of Bcl-2/Bax, the release of cytochrome c from the mitochondria, and the subsequent activation of Apaf-1/caspase-9 apoptosome. The presence of PARP proteolysis further substantiates the evidence of apoptosis in these cells, emphasizing the significant role of cytochrome c–initiated caspase cascade in the mitochondrial pathway of apoptosis triggered by glutamate exposure. However, glutamate-induced mitochondria-dependent cascades were attenuated in HT-22 cells treated with γ-oryzanol. γ-Oryzanol protects neurons from glutamate excitotoxicity by reducing intracellular ROS and inhibiting mitochondrial apoptotic signaling, which could thus be considerable.

Several studies have documented various potential molecular mechanisms underlying the neurotoxicity induced by excessive glutamate, with some suggesting that this phenomenon could be facilitated through the activation of MAPKs [6]. Excessive activation of NMDA receptors by glutamate can trigger the JNK–c-Jun pathway, leading to cerebral ischemia/reperfusion injury and neuronal apoptosis [10]. The phosphorylation of c-Jun by JNKs and subsequent AP-1 transcriptional activity enhances the expression of pro-apoptotic genes such as Bax, ultimately leading to apoptotic cell death [10]. Studies have shown that ASK1 may be a key element in the upstream MAPK signaling cascade that can activate MKK4 or MMK7 to initiate a JNK-mediated apoptotic pathway in a cell-type and stimuli-specific manner [9]. Our research has found that exposing HT-22 cells to glutamate injury leads to increased levels of phosphorylation not only for JNK but also for its activators, which include ASK1, MKK4, and MKK7; additionally, the downstream target activation of c-Jun and c-Fos, the members of the AP-1 complex, was also found to be elevated. The involvement of MAPK pathways, particularly the ASK1/MKK4/7/JNK/AP-1 signaling cascade, has been reasonably implicated in glutamate-induced apoptosis in HT-22 cells. After being exposed to glutamate, treatment with γ-oryzanol reduced ASK1 and MKK4/7 phosphorylation; consequently, this interference with the JNK signaling pathway decreased the activation of downstream targets of c-Jun/c-Fos. It is not yet sure whether γ-oryzanol directly or indirectly blocks MAPK-dependent neuronal apoptosis; however, it inhibits MAPK signal transduction pathways similarly to the effect produced by memantine, which binds and non-competitively inhibits NMDA receptors [39]; therefore, the ASK1/MKK4/7/JNK/AP-1 axis seems to play a crucial role in the ability of γ-oryzanol to counter glutamate-induced apoptosis in HT-22 hippocampal nerve cells.

It is important to note that the presence of glutamate triggers the activation of NMDA receptors, leading to an influx of calcium that affects several downstream signaling pathways. Among these, CaMKII plays a significant role in regulating neuroreceptors and initiating a series of apoptotic signaling pathways that depend on ASK1/JNK [40]. γ-Oryzanol significantly reduced calcium influx and CaMKII activation, broadening our understanding of its potential to block the NMDA receptor/CaMKII signaling cascade in HT-22 cells undergoing glutamate excitotoxicity. Memantine, functioning as an NMDA receptor antagonist, enhances memory by restoring homeostasis in the glutamatergic system [21]. In patients with mild to moderate vascular dementia, a daily dosage of 20 mg of memantine consistently improved cognition across various cognitive scales [41]. Our study, utilizing an in vitro cell model of differentiated HT-22 cells under glutamate-induced excitotoxicity, is the first to demonstrate the potential therapeutic effects of γ-oryzanol on vascular dementia, comparable to memantine, despite its established neuroprotective properties. Through research on treatments for CNS diseases, many ineffective drugs have been produced, with most of them being discarded due to their inability to cross the BBB [42]. Although the BBB permeability of γ-oryzanol can be considered as evidence supporting its therapeutic effects on glutamate-induced neurodegeneration [14], further research is required to verify these findings through animal models or human studies.

## 5. Conclusions

In conclusion, the current experiment has confirmed that γ-oryzanol has potent neuroprotective properties against glutamate excitotoxicity in differentiated HT-22 cells. The compound helps to reduce oxidative stress, prevent the loss of mitochondrial membrane potential, and lower calcium overload. Additionally, it prevents glutamate-induced cell apoptosis by decreasing CaMKII activation to block the ASK-1/c-Jun/AP-1 cascade. These findings suggest that γ-oryzanol could be a promising candidate for the development of therapies for neurodegenerative diseases associated with glutamate excitotoxicity.

## Figures and Tables

**Figure 1 nutrients-16-01237-f001:**
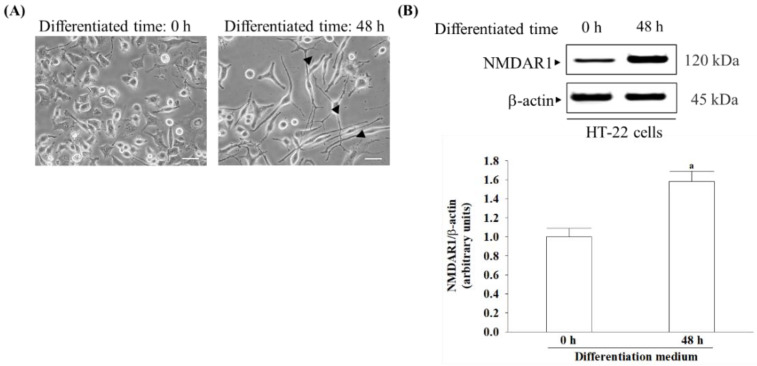
Differentiation transformed into a mature neuron and expressed the NMDA receptor in HT-22 cells. (**A**) HT-22 cell morphology observed before differentiation (left panel) and after 48 h in a differentiation medium (right panel). After 48 h of culture in differentiation medium, HT-22 cells developed a neuron-like morphology, characterized by the presence of extended neurites. Black arrows indicate neurites. Original magnification: 200×; scale bar: 50 μm. (**B**) Differentiation conditions induce the expression of the NMDA receptor in HT-22 cells after 48 h. NMDA receptor subunit 1 (NMDAR1) was detected as the NMDA receptor. β-actin was used as loading control. The results are shown as the mean ± SD of five independent experiments (*n* = 5) performed in triplicate. ^a^
*p* < 0.05 compared to the undifferentiated group.

**Figure 2 nutrients-16-01237-f002:**
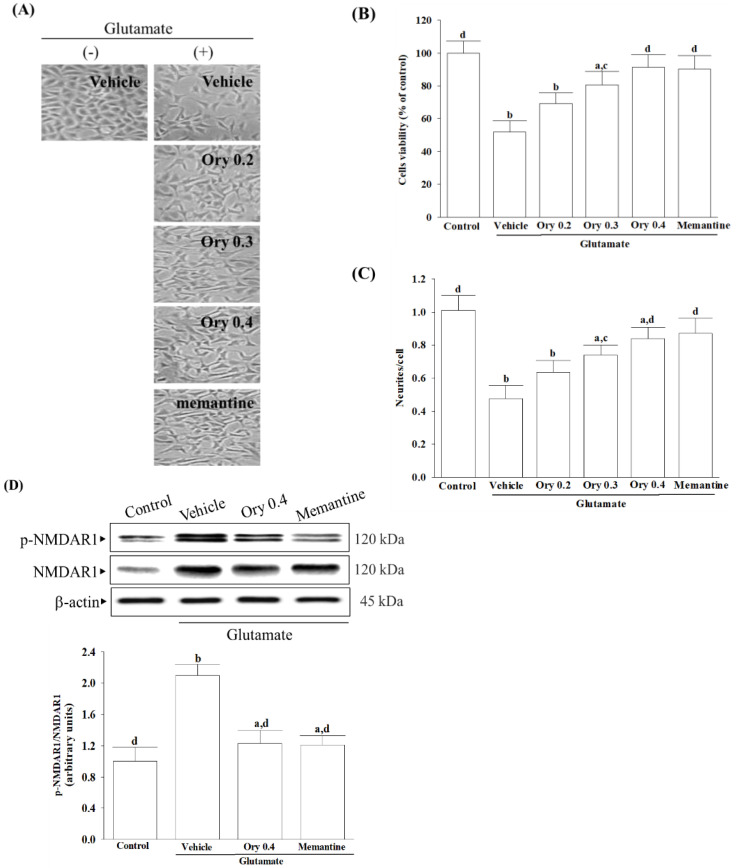
γ-Oryzanol alleviates glutamate-induced neuronal damage. (**A**) Cell morphology, (**B**) cell viability, (**C**) neurite outgrowth, and (**D**) protein levels and phosphorylation degree of NMDA receptor were evaluated when the differentiated HT-22 cells were treated with 0.1 mmol/L glutamate for 24 h and received γ-oryzanol (Ory) at 0.2 (Ory 0.2), 0.3 (Ory 0.3), and 0.4 (Ory 0.4) mmol/L or memantine (10 µmol/L) for another 24 h. The cell morphology was examined using an Zeiss Axiovert 135 inverted phase-contrast microscope at ×100 magnification. Cell viability was determined with a CCK-8 assay and expressed as a percentage of untreated cells taken as a control group. The total number of neurites per cell was determined by counting the number of neurites directly from the soma using Image J software (version 1.6.0). Protein levels and phosphorylation degree of NMDAR1 are shown via representative immunoblots. β-actin was used as loading control. The ratio between phosphoprotein and total protein of NMDAR1 (p-NMDAR1 and NMDAR1) was calculated. The results are shown as the mean ± SD of five independent experiments (*n* = 5) performed in triplicate. ^a^ *p* < 0.05 and ^b^ *p* < 0.01 compared to the untreated control group (control). ^c^ *p* < 0.05 and ^d^ *p* < 0.01 compared to the data from glutamate-stimulated cells that received vehicle treatment.

**Figure 3 nutrients-16-01237-f003:**
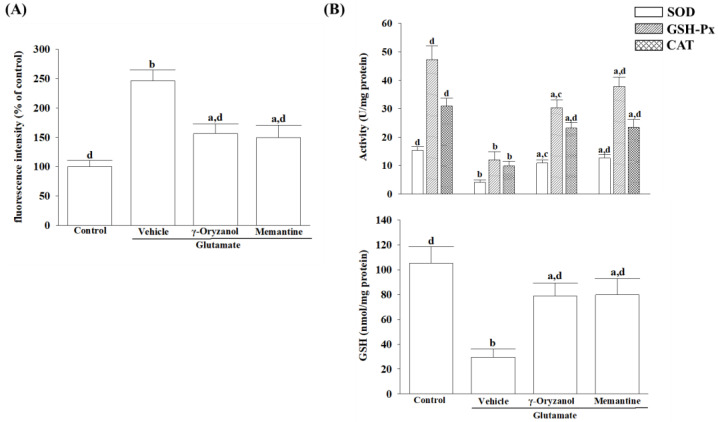
γ-Oryzanol protects differentiated HT-22 cells against glutamate-induced oxidative stress. The differentiated HT-22 cells were treated with 0.1 mmol/L glutamate for 24 h and received γ-oryzanol (0.4 mmol/L) or memantine (10 µmol/L) for another 24 h. (**A**) The ROS fluorescence intensity is expressed as a percentage of the untreated control cells. (**B**) The activities of SOD, GSH-Px, CAT, and GSH content were normalized to the corresponding protein concentration for each group. The results are shown as the mean ± SD of five independent experiments (*n* = 5) performed in triplicate. ^a^ *p* < 0.05 and ^b^ *p* < 0.01 compared to the untreated control group (control). ^c^ *p* < 0.05 and ^d^ *p* < 0.01 compared to the data from glutamate-stimulated cells that received vehicle treatment.

**Figure 4 nutrients-16-01237-f004:**
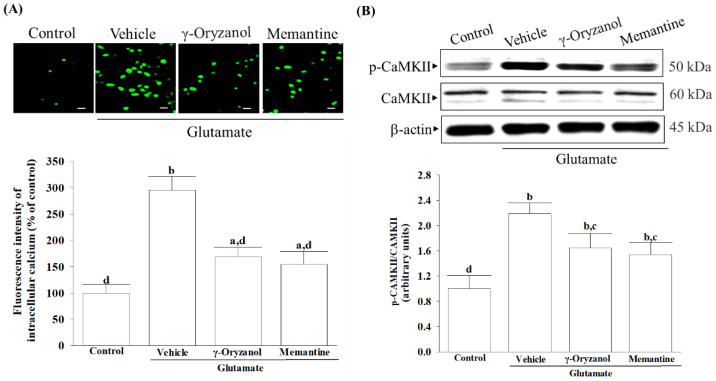
γ-Oryzanol suppressed glutamate-induced intracellular calcium overload and CaMKII activation. The differentiated HT-22 cells were treated with 0.1 mmol/L glutamate for 24 h and received γ-oryzanol (0.4 µmol/L) or memantine (10 µmol/L) for another 24 h. (**A**) The fluorescence intensities for intracellular Ca^2+^ was measured via live cell imaging. Original magnification: 200×; scale bar: 50 μm. Represents bar diagram of relative fluorescence intensity. (**B**) Representative immunoblots depicting the protein levels and phosphorylation degree of CaMKII, β-actin was used as loading control. The ratio of p-CaMKII to total CaMKII was calculated. The results are shown as the mean ± SD of five independent experiments (*n* = 5) performed in triplicate. ^a^ *p* < 0.05 and ^b^ *p* < 0.01 compared to the untreated control group (control). ^c^ *p* < 0.05 and ^d^ *p* < 0.01 compared to the data from glutamate-stimulated cells that received vehicle treatment.

**Figure 5 nutrients-16-01237-f005:**
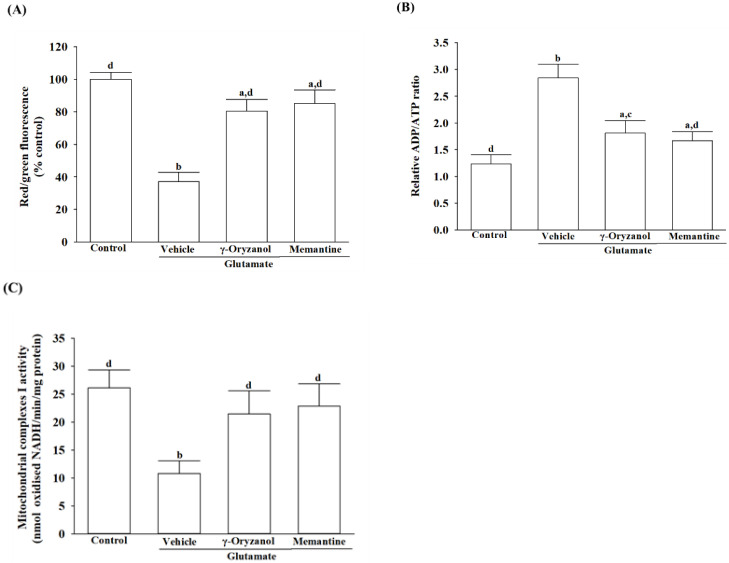
γ-Oryzanol prevents glutamate-induced mitochondrial dysfunction. The differentiated HT-22 cells were treated with 0.1 mmol/L glutamate for 24 h and received γ-oryzanol (0.4 µmol/L) or memantine (10 µmol/L) for another 24 h. (**A**) The mitochondrial transmembrane potential has been measured with the JC-1 fluorescence probe. (**B**) The bioluminescent detection of ADP and ATP levels was used to measure the ADP/ATP ratio in cells with a commercial assay kit. (**C**) The oxidation of NADH was measured to determine the activity of mitochondrial complex I. The results are shown as the mean ± SD of five independent experiments (*n* = 5) performed in triplicate. ^a^ *p* < 0.05 and ^b^ *p* < 0.01 compared to the untreated control group (control). ^c^ *p* < 0.05 and ^d^ *p* < 0.01 compared to the data from glutamate-stimulated cells that received vehicle treatment.

**Figure 6 nutrients-16-01237-f006:**
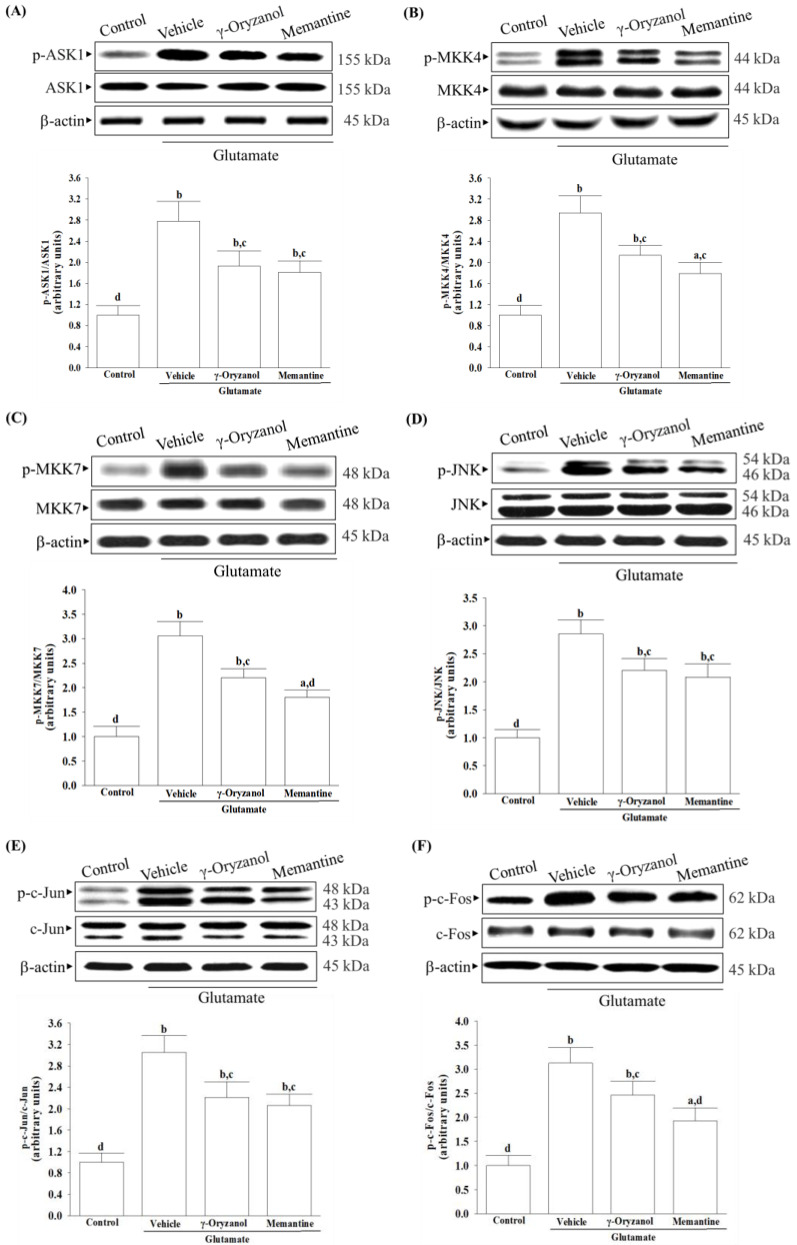
γ-Oryzanol downregulates ASK1/JNK signaling pathway induced by glutamate. The differentiated HT-22 cells were treated with 0.1 mmol/L glutamate for 24 h and received γ-oryzanol (0.4 µmol/L) or memantine (10 µmol/L) for another 24 h. Protein expression and extent of phosphorylation on (**A**) ASK1, (**B**) MKK4, (**C**) MKK7, (**D**) JNK, (**E**) c-Jun, and (**F**) c-Fos were analyzed by Western blot. The degree of phosphorylation was calculated as a ratio of the total protein. β-actin was used as loading control. The results are shown as the mean ± SD of five independent experiments (*n* = 5) performed in triplicate. ^a^
*p* < 0.05 and ^b^
*p* < 0.01 compared to the untreated control group (control). ^c^
*p* < 0.05 and ^d^
*p* < 0.01 compared to the data from glutamate-stimulated cells that received vehicle treatment.

**Figure 7 nutrients-16-01237-f007:**
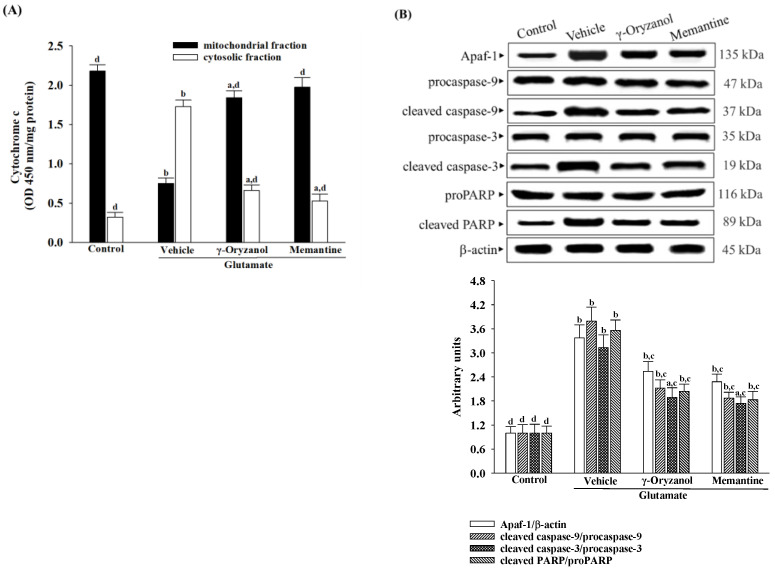
γ-Oryzanol declined cytochrome c-initiated caspase cascade activation induced by glutamate. The differentiated HT-22 cells were treated with 0.1 mmol/L glutamate for 24 h and received γ-oryzanol (0.4 µmol/L) or memantine (10 µmol/L) for another 24 h. (**A**) The concentrations of cytochrome c in both mitochondrial and cytosolic fractions were determined using an immunoassay. (**B**) Protein expression of Apaf-1, procaspase-9, cleaved caspase-9, procaspase-3, cleaved caspase-3, proPARP, and cleaved PARP were analyzed by Western blot. Β-actin was used as loading control. The results are shown as the mean ± SD of five independent experiments (*n* = 5) performed in triplicate. ^a^
*p* < 0.05 and ^b^
*p* < 0.01 compared to the untreated control group (control). ^c^
*p* < 0.05 and ^d^
*p* < 0.01 compared to the data from glutamate-stimulated cells that received vehicle treatment.

**Figure 8 nutrients-16-01237-f008:**
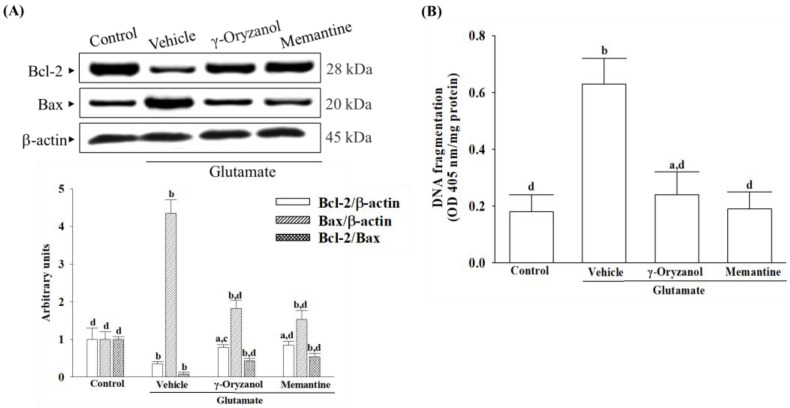
γ-Oryzanol reversed Bcl-2 protein downregulation and DNA fragmentation induced by glutamate. The differentiated HT-22 cells were treated with 0.1 mmol/L glutamate for 24 h and received γ-oryzanol (0.4 µmol/L) or memantine (10 µmol/L) for another 24 h. (**A**) The expression of Bcl-2 and Bax protein were measured by Western blot. β-actin was used as loading control. The ratio of relative intensities in Bcl-2 to Bax (Bcl-2/Bax) was reported. (**B**) The extent of apoptosis was quantified using the ELISA kit to detect DNA fragments associated with cytoplasmic histones. The results are shown as the mean ± SD of five independent experiments (*n* = 5) performed in triplicate. ^a^
*p* < 0.05 and ^b^
*p* < 0.01 compared to the untreated control group (control). ^c^
*p* < 0.05 and ^d^
*p* < 0.01 compared to the data from glutamate-stimulated cells that received vehicle treatment.

## Data Availability

All the data needed to evaluate the conclusions in the paper are present in the paper. Additional data related to this paper may be requested from the authors.

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
