# Peer review of "γ-Oryzanol from Rice Bran Antagonizes Glutamate-Induced Excitotoxicity in an In Vitro Model of Differentiated HT-22 Cells"

_nutrients, 2024, doi:10.3390/nu16081237_

Round 1

Reviewer 1 Report

Comments and Suggestions for Authors

Dear Editor,

thank you for inviting me to review the Manuscript ID: nutrients-2850284 entitled: “γ-Oryzanol from Rice Bran Antagonizes Glutamate-Induced Excitotoxicity in an in Vitro Model of Differenced HT-22 Cells”, authors: Li-Chai Chen, Mei Chou Lai, Tang-Yao Hong, I-Min Liu, submitted to the special issue “The Role of Micronutrients in Neurodegenerative Disease” of Nutrients.

This study aimed to assess the effect of γ-Oryzanol, a natural compound that 
enhances brain function,
on glutamate induced excitotoxicity to explore its neuroprotective properties, using an in vitro model of differentiated HT-22 cells under glutamate-induced damages. The Authors suggest that γ-oryzanol has potent neuroprotective properties against glutamate excitotoxicity.

The work was well designed, however, revisions are needed to clarify issues that do not make it possible to fully understand the results, as detailed below.

General comments:

- Abstract section:

The sentence “This study investigated if γ-oryzanol can delay or improve vascular dementia” refers to a biological relevance of γ-Oryzanol regarding its potential effect that is not determined in this study, therefore this sentence must be deleted or needs to be changed to a more appropriate one.

The definition “an in vivo model of differentiated HT-22 cells” must be changed with “an in vitro model of differentiated HT-22 cells” as in the title of Manuscript.

- The Result section is poor and it must be improved to better illustrate the results obtained.

To make the results homogeneous and understandable, in all analyzes carried out, the Authors must decide whether to report the effects of γ-Oryzanol as a percentage or as a fold change vs control group, and standardize all the graphs.

- Western blotting analysis: in all the figures, the molecular mass of each signals is not showed, this information is required because the whole immunoblot image is not showed, therefore the Authors must add this value.

- The English language is poor and need improvement; typo errors must be corrected.

Specific comments

1. The structure of γ-Oryzanol must be shown and its chemical and biological properties better described.

2. Figure 1:

- panel A shows HT-22 cell morphology observed before differentiation (1A) (this time is usually rereferred as Time 0) and after 48 h in a differentiation medium (1A) (in the results the relative image position, i.e. left or right, is not specified). In this panel, undifferentiated cells at 48 must be showed to compare time-dependent effect on cell proliferation and morphology. In addition, each image must be labeled with the relative indication of the observation time and culture media conditions, in the result description and figure legend.

- In the Materials and Methods, section 2.1.

- Line 117: the sentence “The total number of neurites per cell was determined by counting the number of neurites directly stemming from the soma using the software image J…….” however, this quantitative analysis is not showed in the figure, therefore this data must be added in this figure and results section.

- Line 122: The sentence “Differentiated HT-22 cells were seeded in 6-well plates at a density of 1 × 10 5 cells per well …. is ambiguous, since the cells were differentiated after the plating.

- panel B, molecular mass of the signals is not showed, this information is required because the whole immunoblot image is not showed.   

- last panel, the bar graph needs to be indicated with a letter; on the y-axis NMDAR1 protein fold change vs untreated cells should be specified, since it appear that the value corresponds to 1 in undifferentiated cells.

3. Materials and Methods

In this section, the topic regarding the in vitro cell model and treatments, in particular the group of control cells used for the quantification of the effect induced by γ-Oryzanol is extremely confused. - - Line 131: “Cells that were not subjected to any treatment and cultured under normal conditions were used as control samples”.

- Line 144: ”The control cells treated with the vehicle alone were considered 100% viable, and the cell viability of the treated samples was expressed as a percentage relative to the control

To make the results homogeneous and understandable, in all analyzes carried out, the following control groups must be shown:

a. undifferentiated cells

b. differentiated cells

c. differentiated cells treated with the vehicle alone (DMSO below 0.1 % line 136, please, specify the exact final concentration) in absence of glutamate.

For quantitative analysis of γ-Oryzanol’s effect (on glutamate-induced neuronal damages (Figure 2), oxidative stress (Figure 3), intracellular calcium overloaded and CaMKII activation (Figure 4), mitochondrial dysfunction (Figure 5), ASK1/JNK signaling pathway (Figure 6), Cytochrome c-initiated caspase cascade (Figure 7), Bcl-2 protein downregulation and DNA fragmentation (Figure 8) the control group set as 100 % must be represented by group c.

4. Results section 3.2.  γ-Oryzanol alleviates glutamate-induced neuronal insults

Figure 2: add cell morphological observation by phase contrast microscopy to show the effect of treatments on cell viability, differentiation and neurites outgrowth, that has been quantify in the panels A and B.

Line 99: “… in vivo model…” change with in vitro model

Line 170: add a reference for Bradford protein assay.

Line 213 and 223: Biorad is a commercial company, change with the method used.

Line 241: add the Tween final concentration.

Line 256: the sentence “expressed in connection with the β-actin”, is not clear, add a more appropriate sentence.

Line 286: the sentence “γ-Oryzanol was found to have a concentration-based elimination effect on the decrease in viability…” must be changed with more appropriate sentence.

Line 305: “…with β-actin as positive control…”. β-actin is not a positive control, change with more appropriate term.

Line 311: allivated ???

I believe that this Manuscript is promising, however, it can only be reconsidered for publication, after appropriate revisions and improvement of the English language.

Comments on the Quality of English Language

The Manuscript needs improvement of the English language

Author Response

Please find it in the attached file.

Reviewer 2 Report

Comments and Suggestions for Authors

Dear authors,

The present study reports the neuroprotective effect of gamma-oryzanol, a natural compound found in rice, in an in vitro neuronal model using differentiated HT22. The work highlights the neuroprotective potential against glutamate-induced excitotoxicity in vitro.

The novelty and interest of the proposed work is not high enough for publication in a Q1 journal since the authors do not demonstrated a therapeutic potential on vascular dementia (Line 558) and the antioxidant and antiapoptotic mechanisms of the compound is already well-known. In addition, the model employed is very simple to state the previous comments.

Some specific comments include:

-         - In the introduction section, the underlying vascular risk factors of vascular dementia should be mentioned.

-         - Line 15 and 99 change in vivo for in vitro

-          The design of figure panels should be improved, especially for figure 6

-          For western blot figures, the specific molecular weight of the ladder next to each protein should be included in the side of the image.

-         - For all graphics, individual values should be represented instead of using bars.

-         - For figure 2b, representative images would be helpful

-         - Line 286 and 287 are difficult to understand for the reader and should be rephrased

-          -Why the NMDA inhibitor memantine is not protective agind excitotoxicity-induced cell death?

-         - Figure 4a. The quality of the images is very low.

-          -The difference between this compound and the reference compound memantine should be discussed in the discussion section.

Comments on the Quality of English Language

In the title: differenced should be changed by differentiated

Author Response

Please find it in the attached file.

Round 2

Reviewer 1 Report

Comments and Suggestions for Authors

Dear Editor,

Thank you for inviting me to review the revised version of the Manuscript ID: nutrients-2850284-peer-review-v2 entitled: “γ-Oryzanol from Rice Bran Antagonizes Glutamate-Induced Excitotoxicity in an in Vitro Model of Differenced HT-22 Cells”, authors: Li-Chai Chen, Mei Chou Lai, Tang-Yao Hong, I-Min Liu, submitted to the special issue “The Role of Micronutrients in Neurodegenerative Disease” of Nutrients, Section: Micronutrients and Human Health.

In this revised manuscript, text and figures have been modified and some additional experimental results have been added according to the previous reviewers’ comments.

I have no further comments.

Comments on the Quality of English Language

The quality of English language meets the international standard of scientific written communications.

Reviewer 2 Report

Comments and Suggestions for Authors

Dear authors,

despite the quality of the paper has been improved by incorporating reviewers´s suggestion, there are still many things that should be improved, such as the quality of the images in figure 2 and 4 or the design of the figure panels.

However, the scientific quality and impact of the manuscript not reach the level needed to be published.